META-RESEARCH ARTICLE

# Blind spots on western blots: Assessment of common problems in western blot figures and methods reporting with recommendations to improve them

**Cristina Kroon[1‡], Larissa Breuer[2,3‡], Lydia Jones[4‡], Jeehye An[5,6‡], Ayça Akan[5,7], Elkhansa Ahmed Mohamed Ali[8], Felix Busch[9,10], Marinus Fislage[10], Biswajit Ghosh[3,11], Max Hellrigel-Holderbaum[5,12], Vartan Kazezian[13,14], Alina Koppold[15], Cesar Alberto Moreira Restrepo[4], Nico Riedel[13], Lea Scherschinski[16], Fernando Raúl Urrutia Gonzalez[4,17,18], Tracey L. Weissgerber** [ID][13‡*]

1 Institute of Molecular Biology and Biochemistry, Charité–Universitätsmedizin Berlin, Berlin, Germany, 2 German Center for Neurodegenerative Diseases (DZNE) within the Helmholtz Association, Berlin, Germany, 3 Department of Biology, Chemistry, Pharmacy, Freie Universität Berlin, Berlin, Germany, 4 Berlin School of Public Health, Charité–Universitätsmedizin Berlin, Berlin, Germany, 5 Berlin School of Mind and Brain, Humboldt-Universität zu Berlin, Berlin, Germany, 6 Department of Experimental Neurology and Center for Stroke Research, Charité–Universitätsmedizin Berlin, Berlin, Germany, 7 Max Planck Institute for Human Cognitive and Brain Sciences, Leipzig, Germany, 8 Department of Psychiatry and Neurosciences, Charité–Universitätsmedizin Berlin, Berlin, Germany, 9 Department of Radiology, Charité–Universitätsmedizin Berlin, Berlin, Germany, 10 Department of Anesthesiology and Intensive Care Medicine, Charité–Universitätsmedizin Berlin, Berlin, Germany, 11 Max Planck Institute for Molecular Genetics, Berlin, Germany, 12 Interdisciplinary Center of Sleep Medicine, Charité–Universitätsmedizin Berlin, Berlin, Germany, 13 QUEST Center for Responsible Research, Berlin Institute of Health at Charité—Universitätsmedizin Berlin, Berlin, Germany, 14 Department of Biochemistry, Freie Universität Berlin, Berlin, Germany, 15 Institute for Systems Neuroscience, University Medical Center Hamburg-Eppendorf, Hamburg, Germany, 16 Department of Neurosurgery, Charité–Universitätsmedizin Berlin, Berlin, Germany, 17 Institute of Public Health, Charité–Universitätsmedizin Berlin, Berlin, Germany, 18 Center for Stroke Research Berlin, Charité–Universitätsmedizin Berlin, Berlin, Germany

‡ CK, LB, LJ, and JA were recognized by their co-authors for leading contributions to the project and are ranked in order of contribution. TLW was the course instructor. All other authors are listed alphabetically.
* tracey.weissgerber@bih-charite.de

**Data Availability Statement:** The abstraction protocols, search strategy, data, data visualization code and teaching slides are deposited in the Open

## Abstract

Western blotting is a standard laboratory method used to detect proteins and assess their expression levels. Unfortunately, poor western blot image display practices and a lack of detailed methods reporting can limit a reader's ability to evaluate or reproduce western blot results. While several groups have studied the prevalence of image manipulation or provided recommendations for improving western blotting, data on the prevalence of common publication practices are scarce. We systematically examined 551 articles published in the top 25% of journals in neurosciences ($n = 151$) and cell biology ($n = 400$) that contained western blot images, focusing on practices that may omit important information. Our data show that most published western blots are cropped and blot source data are not made available to readers in the supplement. Publishing blots with visible molecular weight markers is rare, and many blots additionally lack molecular weight labels. Western blot methods sections often lack information on the amount of protein loaded on the gel, blocking steps, and antibody labeling protocol. Important antibody identifiers like company or supplier,

Science Framework (OSF) repository (RRID: SCR_003238) at https://osf.io/2c4wq/.

**Funding:** This study was completed through a participant guided, learn-by doing meta-research course funded by the Berlin University Alliance within the Excellence Strategy of the federal and state governments (301_TrainIndik, TLW). The funders had no role in study design, data collection and analysis, decision to publish, or preparation of the manuscript.

**Competing interests:** The authors have declared the no competing interests exist.

catalog number, or RRID were omitted frequently for primary antibodies and regularly for secondary antibodies. We present detailed descriptions and visual examples to help scientists, peer reviewers, and editors to publish more informative western blot figures and methods. Additional resources include a toolbox to help scientists produce more reproducible western blot data, teaching slides in English and Spanish, and an antibody reporting template.

## Introduction

Western blotting or immunoblotting is a common laboratory method used to detect proteins and assess their expression levels. A protein of interest is identified based on its molecular weight and immunoreactivity with a specific antibody. Western blotting consists of a series of interrelated steps (Fig 1). Small variations in how these steps are performed can alter the quality of the blot, introduce errors, or affect the interpretation of experimental results [1]. This has contributed to concerns about the reproducibility and reliability of western blot experiments [2–6]. Although publication practices are just 1 factor contributing to skepticism around western blotting, improving these practices could have an important impact on reproducibility and trustworthiness. Informative figures and detailed methods sections help readers to identify well-executed experiments and potential sources of error, while providing information needed to replicate the experiment.

Fig 2 highlights several western blot image display practices that can omit information necessary to interpret western blots, like narrowly cropped blots to display only the band of interest [7–11], omitted molecular weight markers [9,10], and missing or poorly used molecular weight labels [8,10,11]. Digital cropping to display only the bands of interest can save space, allow authors to efficiently combine many blots into a single figure, and focus readers' attention on the band of interest. However, these benefits deprive readers of essential information. Full-length blots displaying the entire vertical length of the gel (Fig 2.1) provide important information about protein multiplicity or antibody specificity by allowing readers to see whether additional bands are present [7,12]. The molecular weight marker, annotated with molecular weight labels, serves as a scale bar to confirm that the detected protein is of the expected size (Fig 2).

In some publications, western blots are quantified to assess a target protein's expression levels. The optical density of protein bands is determined based on their area and intensity [6]. Many factors limit accurate, reproducible western blot quantification (see, for example, [4,6,13,14]); therefore, the method is considered to be only semiquantitative. The sample size in western blot experiments is typically small; however, quantification data are often presented in bar graphs. This leads to loss of transparency, as many different datasets can lead to the same bar graph and the individual data points may suggest different conclusions from the summary statistics alone [15].

Details of the western blotting method, such as the amount of total protein loaded onto the gel (Fig 1.1) [16], the membrane blocking and antibody incubation protocols, and the antibodies used (Fig 1.4), heavily influence the quality of the blot. Small changes in these factors can dramatically alter the level of background, amount of nonspecific binding, and intensity of the bands [1,9,17]. Many authors have acknowledged the need to improve reporting for each of these items [4,11]. Antibody selection is one of the most critical factors, since nonspecific antibodies are a well-documented cause of irreproducible or incorrect western blot results [4,18–

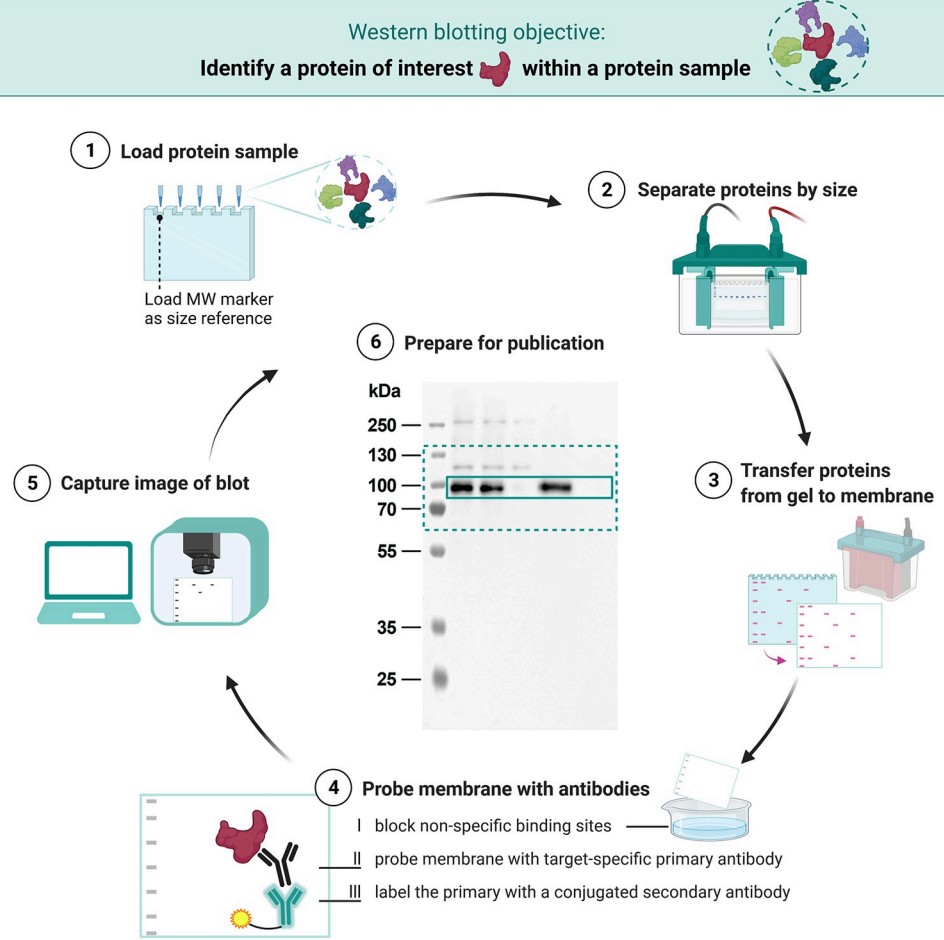

**Fig 1. Western blot: From gel to publication.** Western blotting is a standard laboratory method that uses antibodies to detect target proteins in a sample. (1) The sample, typically a mixture of proteins, is loaded on the gel. A molecular weight (MW) marker, which contains prelabeled proteins of varied, known molecular weights, is loaded on the gel alongside the protein sample as a size reference. (2) Gel electrophoresis is used to separate proteins based on their molecular weight. (3) The proteins are transferred, or "blotted", onto a membrane. (4) The membrane is blocked to reduce nonspecific binding and then sequentially probed with a primary antibody that specifically binds to the protein of interest and a secondary antibody. The latter binds the primary antibody and carries an enzyme or a fluorophore that allows subsequent detection. (5) The signal is detected through a chemiluminescent reaction or fluorescence, respectively. (6) An image of the western blot is prepared for publication: Annotations are added and often the blot is cropped. For the unprocessed image, see S1 Fig.

21]. One study found that only 44% of the antibodies reported in their sample of biomedical papers could be uniquely identified [22]. Methodological reporting practices of other items have not previously been investigated in multifield studies.

In this study, we systematically reviewed 551 full-length publications in high-impact neurosciences and cell biology journals to assess the prevalence of selected image display, data presentation, and methods reporting practices. We found that in most publications, blots are cropped and lack molecular weight markers. Information on secondary antibodies is more often entirely omitted than information on primary antibodies, which, however, is often incomplete. Lot numbers are rarely reported.

# The benefits of transparency when presenting western blot images

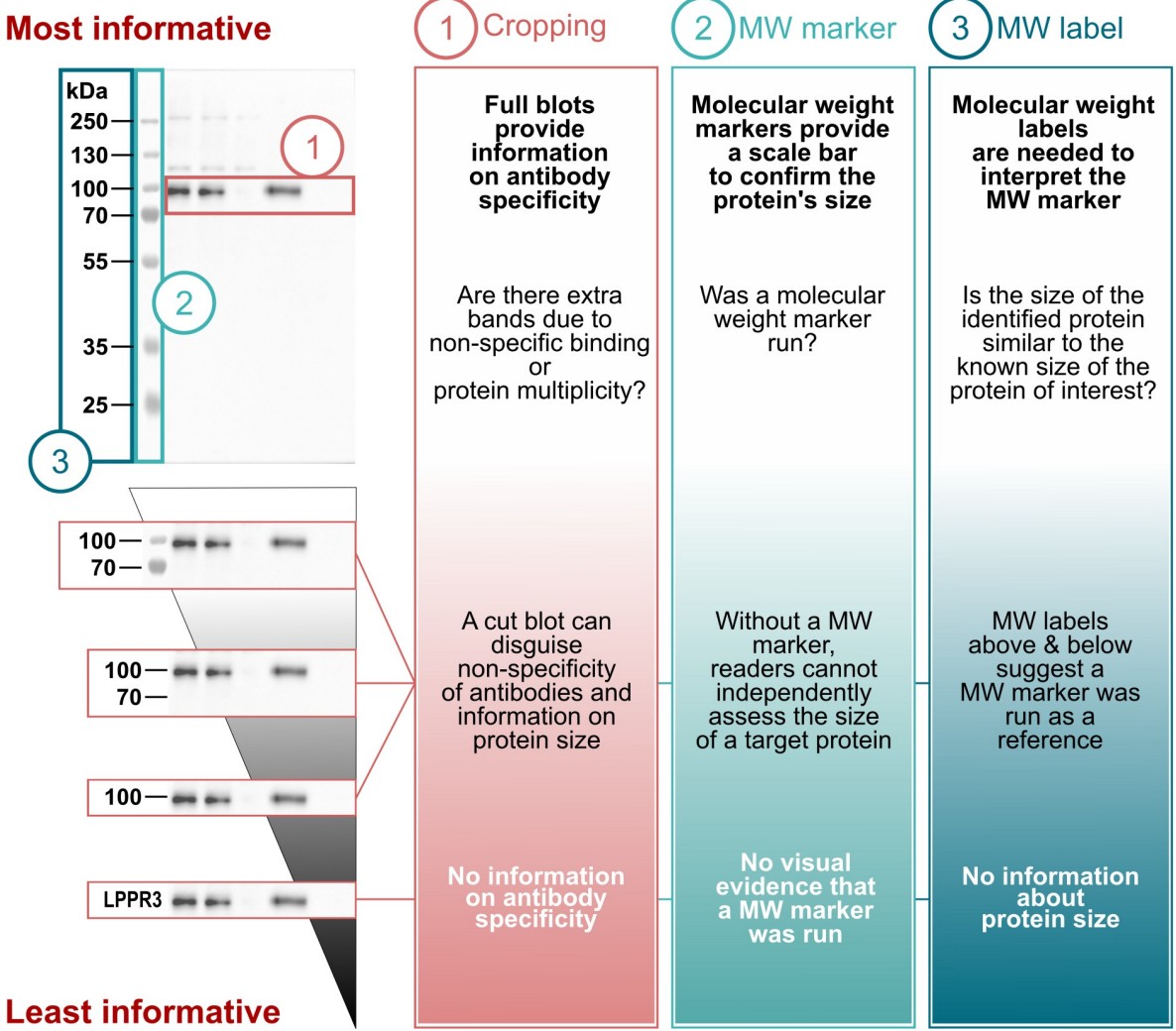

**Fig 2. More and less informative western blot image display practices.** Readers need 3 pieces of visual information to assess western blot results. Suboptimal image display practices can omit this information. (1) Extra bands indicate that the antibody may not be specific to the protein of interest or may recognize multiple forms of the protein of interest. Cropped blots omit this information. (2) Molecular weight (MW) markers allow readers to confirm that protein size was determined using known standards. (3) MW labels show protein sizes, in kDa, for each MW marker band. Displaying the MW marker and labelling MW marker bands above and below the protein of interest is more informative than labelling only the protein of interest. For the unprocessed image, see S1 Fig.

## Results

### Using a science of science approach to investigate current practices

This study was conducted in the context of a participant-guided learn-by-doing course [23] at the Berlin Institute of Health (BIH) at Charité—Universitätsmedizin Berlin. Participants designed and conducted this meta-research project (AA, EA, JA, LB, FB, MF, BG, MH, LJ, AK, CK, CM, LS, and FRUG) together with course instructors (TLW and VK) and a data scientist (NR).

We examined common practices in western blotting image display, data presentation, and methods reporting in 2 fields, neurosciences and cell biology. For each category, we examined

papers in the top 25% of journals that published original research, as defined by impact factor rankings from Journal Citation Reports (JCR; RRID:SCR_017656). All full-length original research articles that contained at least 1 image of a western blot were included in the analysis (S2 Fig). Subsequently, information from each publication was abstracted by 2 independent reviewers (see: Materials and methods section, supplementary materials and supporting information deposited in our OSF repository at https://osf.io/2c4wq/ [24]).

## Figures

We studied western blot image display practices in a sample of 551 full-length papers in neurosciences and cell biology journals that included at least 1 western blot image (Fig 3).

**Cropping.**   We found that cropping blots is common practice in scientific publications. Among the 551 papers that included western blot images, more than 90% had only cropped blots (93% neurosciences; 91% cell biology) (Fig 3A, top panel). Over 80% of the papers did not provide blot source data (unprocessed images) of any cropped blot in the supplement (81% neurosciences; 92% cell biology) (Fig 3A, middle panel). Blots presented as source data had often been vertically cut or cropped: Only 7% of neurosciences and 2% of cell biology papers provided full-length western blot images as supplemental information (Fig 3A, bottom panel).

**Molecular weight markers and labels.**   Publishing blots without a visible molecular weight marker is also common practice. More than 95% of the studied papers lacked molecular weight markers in all western blot images (95% neurosciences; 97% cell biology) (Fig 3B, top panel). Around one-third of the papers in cell biology and neurosciences presented blots without any molecular weight labels (30% neurosciences; 38% cell biology) (Fig 3B, top panel). Approximately a quarter of all papers contained blot figures with only 1 molecular weight label per blot (31% neurosciences; 22% cell biology) (Fig 3B, middle panel). These single molecular weight labels were directly at the protein of interest in approximately a third of all papers (38% neurosciences; 24% cell biology) (Fig 3B, bottom panel). Only 7% of neurosciences and 2% of cell biology papers provided western blot source data with a visible molecular weight marker (Fig 3A, bottom panel).

**Quantification graphs.**   Western blot quantification graphs were less common among papers in cell biology journals (54%) than in neurosciences journals (86%) (Fig 3C). Among these papers, bar graphs were the most common type of graph used to present western blot data (44% neurosciences; 30% cell biology), followed by dot plots (40% neurosciences; 22% cell biology). Other types of graphs, such as line graphs, were much more common in cell biology papers (11%) than neuroscience papers (2%). Large sample size graphs, such as histograms, box plots, or violin plots, were very rare (1% neurosciences; 1% cell biology).

## Methods reporting

We next examined how detailed the descriptions of western blotting methods and materials were in the same dataset (Fig 4).

**Methodological shortcut citations.**   Authors sometimes cite a previous paper that describes the method instead of describing the method in detail themselves. These shortcut citations are used to save space. Almost a third of neurosciences and 20% of cell biology papers used shortcut citations to describe some aspects of western blot methods (Fig 4A).

**Protein loading.**   Approximately 55% of the papers in neurosciences and 78% in cell biology journals did not report the amount of protein loaded on the gel (Fig 4B). Authors who reported this information typically provided exact values (40% neurosciences; 16% cell biology), rather than a range of values (5% to 6% in both fields).

## A  Are the blots in the paper full-length?

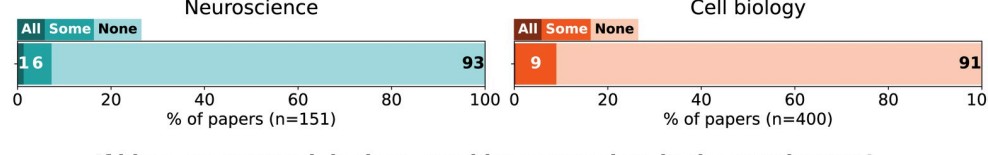

## If blots are cropped, is there any blot source data in the supplement?

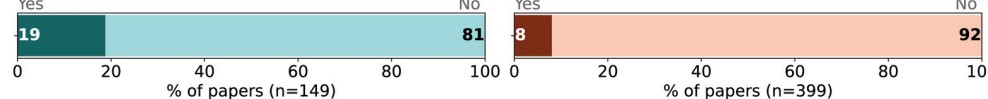

## Are the source data blots…

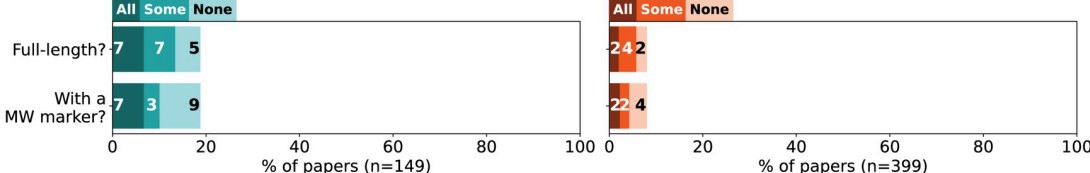

## B  Do blots have MW markers or MW labels?

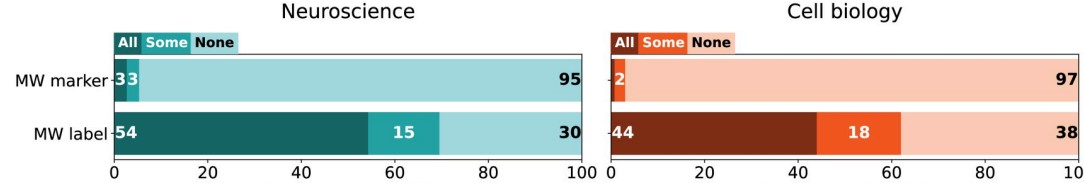

## Do blots with MW label have more than one label?

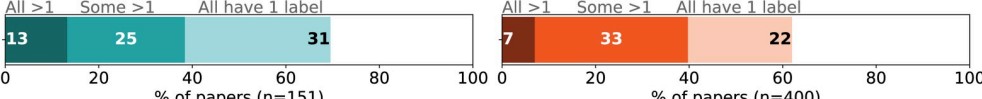

## If blots have one MW label, how is it positioned relative to the protein of interest?

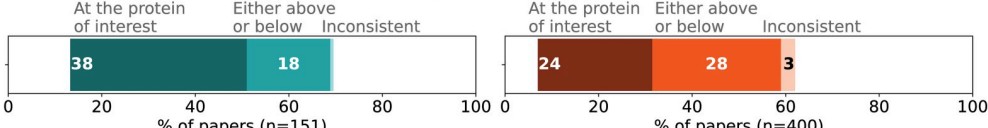

## C  Is there a graph depicting western blot quantification?

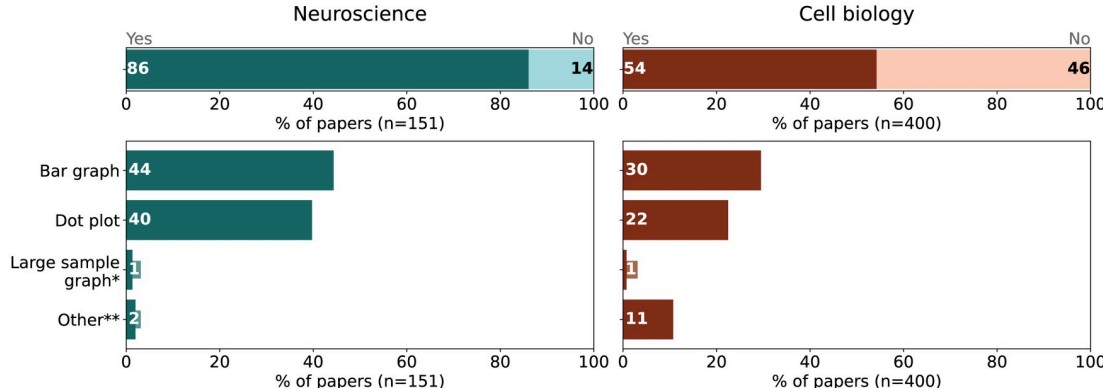

**Fig 3. Western blot figures: Prevalence of image display and data presentation practices that may omit information.** Bar graphs illustrate the prevalence of (A) full-length blots and uncropped blot source data (the bottom panel bar graphs are conditional to the availability of blot source data in the supplement, i.e., middle panel) and (B) molecular weight (MW) markers and labels as well as the position of MW labels (the middle and bottom panel bar graphs are conditional to the presence of MW labels). Bar graphs in (C) illustrate the prevalence of different types of graphs used to visualize western blot quantification data in scientific publications. *Large sample graph—histogram, box plot, violin plot. **Other–graphs that do not fall into previous categories, such as line graphs. Papers may contain multiple types of graphs and thus the sum may exceed the total % of papers with graphs. Totals may not be exactly 100% due to rounding. Labels that annotate <1% are not shown. The data underlying this figure can be found at https://osf.io/2c4wq/ [24].

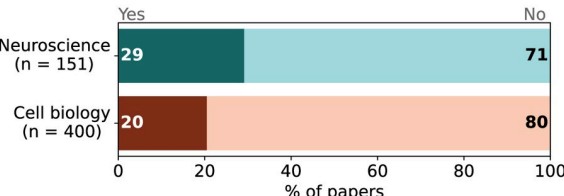

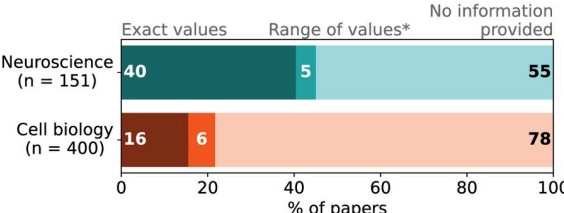

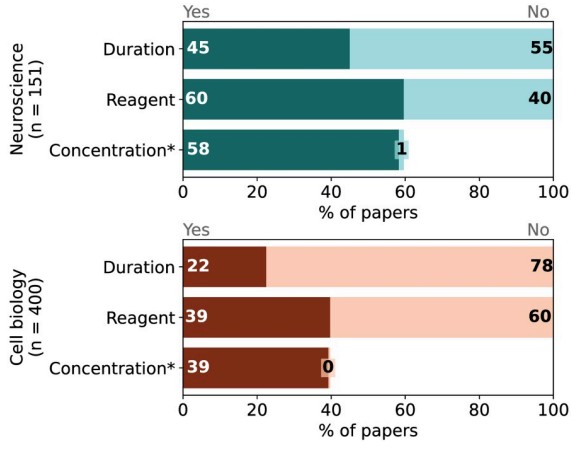

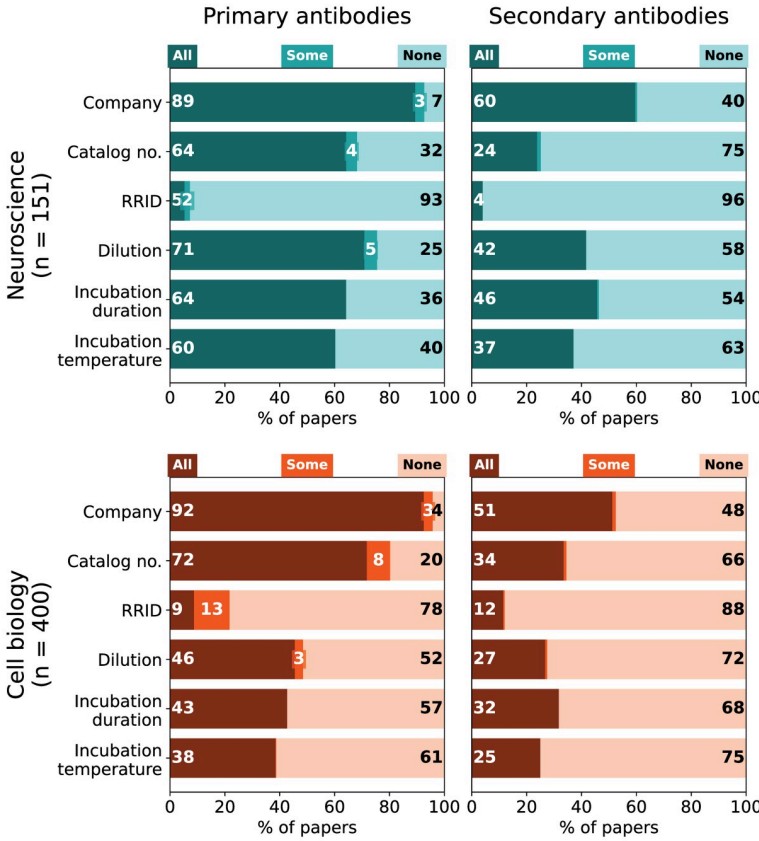

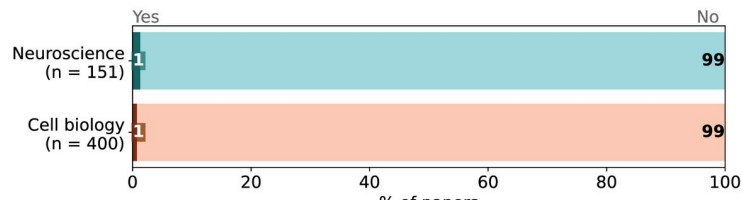

**Fig 4. Western blot methods reporting: Frequency and quality of reporting for selected methods reporting items.** Bar graphs illustrate (A) the use of shortcut citations and the reporting of (B) total protein amount used for blotting. *Fewer than 0.5% of papers reported both exact values and ranges of values; therefore, this category was combined with "Range of values". (C) Blocking step. *Categories All and Some were combined because Some was <1%. (D) Antibody identifiers and labelling protocol. (E) Lot numbers for antibodies. Totals may not be exactly 100% due to rounding. Labels that annotate <2% are not shown. The data underlying this figure can be found at https://osf.io/2c4wq/ [24].

**Membrane blocking protocol.** Details of the membrane blocking protocol were often missing. More than half of the papers in both fields lacked information on blocking duration (55% neurosciences; 78% cell biology) (Fig 4C). Around half of the papers did not report the blocking reagent. Almost all papers that reported the type of blocking reagent also stated the concentration of the blocking reagent.

**Primary and secondary antibodies.** Authors more often reported detailed information on primary antibodies than on secondary antibodies. While only 5% of papers failed to report the company or supplier for all primary antibodies (neurosciences: 7%, cell biology: 4%), the company or supplier for all secondary antibodies was missing in 40% of neurosciences papers and 48% of cell biology papers (Fig 4D). Catalog numbers for primary antibodies were entirely missing in a striking proportion of papers (32% neurosciences, 20% cell biology), although the reporting was again much worse for secondary antibodies where catalog numbers were missing in 75% of neurosciences and 66% of cell biology papers. Research resource identifiers (RRIDs), the most reliable identifiers for antibodies, were rarely reported for primary or secondary antibodies. Only 1% of papers from each field reported a lot number for any antibody (Fig 4E).

Details of the antibody labelling protocol were also missing more often for secondary antibodies than for primary antibodies, and the reporting was generally worse in cell biology papers. The dilution of all antibodies was omitted more frequently for secondary (58% neurosciences; 72% cell biology) than primary (25% neurosciences; 52% cell biology) antibodies (Fig 4D). The incubation time was likewise omitted more frequently for secondary (54% neurosciences; 68% cell biology) than primary (36% neurosciences; 57% cell biology) antibodies. Incubation temperature reporting followed a similar pattern, with 63% of neurosciences and 75% of cell biology papers lacking information for all secondary antibodies and 40% of neurosciences papers and 61% of cell biology for all primary antibodies.

**Replicates.** While the original protocol included a question on reporting of replicates, this question was eliminated as reviewers could not obtain reliable data due to poor reporting quality. Lessons learned from this experience are presented in **Box 1**.

## Discussion

Our systematic assessment of western blot figures and methods in cell biology and neurosciences reveals several prevalent shortcomings and highlights opportunities to improve reporting. The typical western blot figure does not provide the reader with the information needed to assess antibody specificity or confirm that the detected protein is of the expected size. The typical western blot methods section lacks important details needed to replicate the experiment, including the amount of protein loaded onto the gel, the blocking and immunolabeling protocols, and RRIDs and lot numbers for primary and secondary antibodies. The name of the company or supplier and catalog numbers are regularly reported for primary antibodies but are often missing for secondary antibodies. Semiquantified western blot data are often presented in bar graphs, instead of in dot plots that show the sample size and data distribution. We outline a series of recommendations that authors, reviewers, and editors can use to improve western blot figures and methods reporting.

### How can authors make their western blot images more informative?

**Make blot source data available to all readers.** Some journals and guidelines [10,11,26] require authors to submit original uncropped images of blots for peer review, whereas other journals also emphasize the value of sharing the blot source data with readers when a paper is published [27]. Publishing all source data in supplemental files or a public repository allows

Box 1. Strategies for improving reporting of replicates

What causes the confusion in reporting experimental replicates?

Replicates provide evidence that experimental results are reproducible. Reporting the number and type of replicates increases transparency and trustworthiness and allows the readers to assess the generalizability of the findings. Unfortunately, in our experience, the terms used to describe replicates are interpreted differently by researchers in different fields and are often used incorrectly. Genuine replicates, which are sometimes called biological replicates, are independent, provide information about variability between samples, and increase the sample size [25]. An example of this would be measuring the level of a protein of interest in the brain lysates of 5 mice. These terms can be confusing, as many scientists do not consider independent samples to be replicates. Pseudoreplicates, which are sometimes called technical replicates, are not independent and do not increase sample size [25]. They only provide information about measurement errors. An example of this would be measuring the level of a protein of interest in the brain lysate of 1 mouse 5 times. These terms can also be confusing, as the degree of nonindependence required to define something as a technical replicate may differ among labs and fields. Given the potential for misinterpretation and misuse, we encourage authors to clearly describe what they did and avoid using terms like biological replicates, technical replicates, and pseudoreplicates. Authors should concisely explain what samples were measured, and if and how samples were related to each other (independent, matched, samples from different tissues in the same animal, etc.).

Insights gained from examining reporting of replicates for western blots

Our original protocol included a question examining reporting of replicates for western blots; however, reporting quality was too poor to obtain reliable data. Scientists often show 1 representative blot without specifying whether that result was replicated. The western blot methods subsection rarely includes a specific statement about replicates. When replicates are addressed, a general statement is typically found in the figure description or in the "statistical analysis" or "study design" methods subsections. These general statements theoretically apply to a whole figure or paper but do not specify whether they include western blots. Details on what constitutes a replicate are typically not reported.

Common examples of uninformative statements on experimental replicates

**Common statement:** All experiments were independently repeated 3 or more times.

**Problems:** It is not clear whether this applies to western blots. The reader cannot determine the number of replicates for individual experiments, nor what constitutes a replicate. There is no information about missing data or excluded observations or experiments.

**Common statement:** All data from at least 3 independent experiments are presented as mean ± standard deviation.

**Problems:** All problems listed above also apply to this statement. This statement only applies to semiquantified western blots, because summary statistics cannot be calculated without a quantification. Many papers do not include any quantified blots (see Fig 3C).

**Example of a good practice statement:** Western blot experiments in Fig 1A and 1C were replicated 5 times, with each replicate consisting of 1 human brain sample from a different patient. Experiments in Figs 1H and 4F were replicated 3 times using cells from different passages.

**Recommendation:** Share source data for all western blot replicates in your paper's supplement or in a data repository (see discussion).

readers to examine information omitted from cropped blots, helping to improve reproducibility and trust. Sharing data in a public repository is always preferred to sharing data in supplemental files, as data search engines are unlikely to find data stored in supplemental files. Showing original images of all blots is especially important when experimental replicates from western blots are semiquantified. The source data allow the reader to visually examine differences in protein band intensities. Publishing raw data of all replicates confirms that these raw data exist [28] and provides readers with visual confirmation that the blot selected for publication is representative.

Authors can take several steps to make it easier for readers to interpret the source data (Fig 5, right panel). Authors should choose a file structure and naming convention that makes it easy for readers to find the source data that they are interested in. Readers should quickly be able to locate blots corresponding to a particular figure, panel, and protein and identify replicates of each blot. Annotations on the images can further help readers to match raw blot images with corresponding figure panels. Molecular weight markers should be labeled to provide readers with scale information. Finally, authors can highlight the blot and region shown in the figure by placing a red box around the cropped region and stating the figure and panel number next to the box. Annotations should also note lanes that appear on the original blot but were cropped out of the region shown in the figure.

**Include a molecular weight marker and use labels to annotate sizes.** Molecular weight markers, labeled with molecular weights, allow readers to confirm that the detected protein is of the expected size. Some journals state that molecular weight markers must be included on each submitted blot [29]; however, no journal currently explicitly asks authors to include markers in main figure western blot images. The Journal of Biological Chemistry includes markers on blots in one of the examples of western blot crops suitable for publishing [30]. Unfortunately, this is the strongest encouragement in publication guidelines for showing visible molecular weight markers on blot images. Some journals require annotating molecular weight marker positions with labels above and below the protein of interest [30]. We encourage authors to always include a marker, annotated with molecular weight labels, in blot figures (Fig 5, left panel). In addition to providing visual confirmation of protein size, showing marker bands above and below the protein of interest ensures that the size reference was run on the blot, the blot is not too closely cropped and allows readers to see any extra bands of a similar size. In cases where the protein of interest is the same size as a molecular weight marker band, we suggest showing 3 marker bands (at, above, and below the protein of interest). We discourage authors from using single molecular weight labels that are solely based on calculated or

## Western Blot Image Minimal Reporting Standard

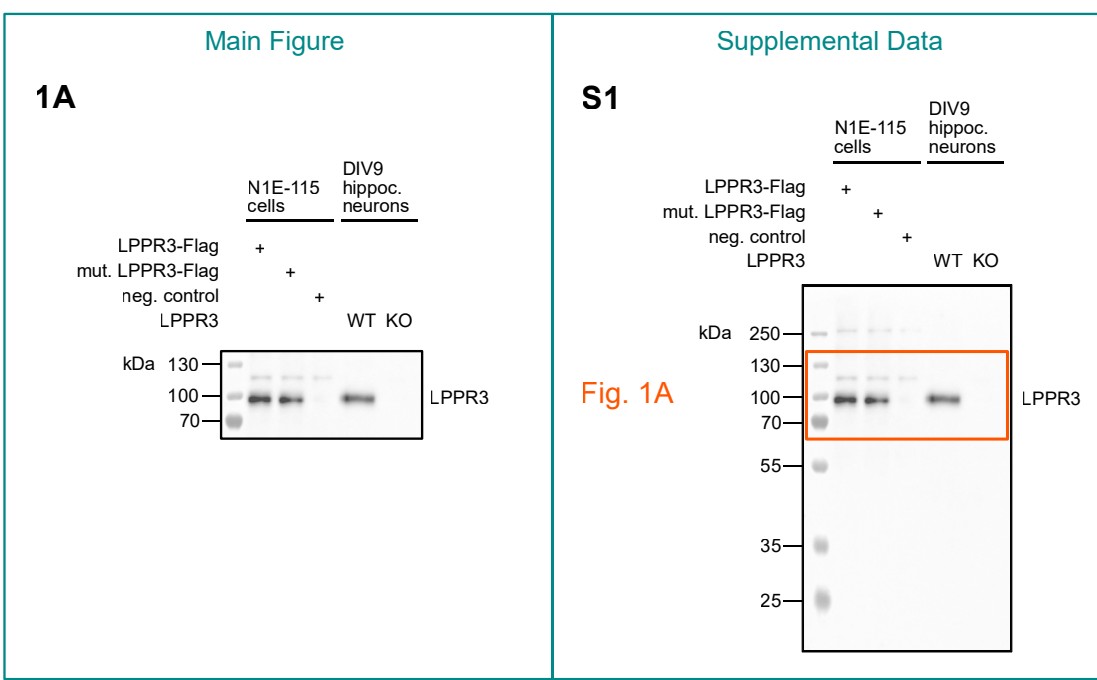

**Fig 5. Western blot image minimal reporting standard.** Published western blots should show a minimum of 2 molecular weight marker bands of different weights if the protein of interest falls between the molecular weight marker bands, and 3 molecular weight marker bands of different weights if the protein of interest is directly at one of the marker bands. The molecular weight markers should be annotated with labels. An original, uncropped image of each blot should be published in the supplement or deposited on a public repository. The source data blot should be named in a way that links it to a specific figure, panel, and protein. The outline of the crop should be annotated on the original image. For the unprocessed image, see S1 Fig.

previously published sizes, as the observed molecular weight of a protein may differ due to many factors [31]. If the observed molecular weight deviates from calculated weight, authors could denote this in the figure description or western blot methods section. Authors can also state the range of the molecular weight marker in the figure description to illustrate how much of the original vertical length is shown.

**Crop your blots as little as possible.** Whenever possible, publish a full-length representative blot. In specific cases, such as antibody validation, blots should always be published as full length. In most other cases, however, space and design considerations can make cropping necessary or even desirable. Some authors have recommended crops to be at least 5 to 6 band widths above and below the protein of interest [7,8]. We support these recommendations with the addition that the crops should also include visible molecular weight markers above and below the protein of interest (Fig 5, left panel). In some laboratories, researchers may physically cut membranes prior to antibody incubation. Some laboratories may do this out of habit, whereas others may want to conserve expensive antibodies. In addition to making it impossible to detect additional bands, this practice also makes it difficult to confirm that controls were run on the same gel. The requirement to provide full blots for every blot relevant to the paper would place labs with limited resources, including those in countries with limited research funding, at a disadvantage. As a compromise, authors could show at least 1 full-length replicate of each individual western blot assay.

**Use dot plots to present western blot quantification data.** Western blot experiments typically include a small number of observations. Bar graphs are the least informative way of presenting continuous data as they conceal the spread of the data and sample size [15,32]. Sample sizes in blot experiments are rarely large enough to justify the use of graphs that only show summary statistics, such as box plots, histograms, or violin plots. We support Pillai-Kastoori and colleagues' recommendation that small sample size western blot data should be presented as dot plots [1]. Similarly, individual data points should be shown in line graphs when illustrating western blot time course experiments. Figure descriptions should clearly identify western blot quantification graphs and connect representative blots to quantification graphs.

## How can authors make their western blot methods section more informative?

A detailed methods section enables readers to replicate the experiment. Our data highlight major deficiencies in reported methods that can have a substantial impact on the outcome of the blot. Most of the methodological details that we assessed are included in the evidence-based list of minimum reporting standards developed by Gilda and colleagues [4]. Being able to uniquely identify the antibodies used in western blots is essential for reproducibility. Scientists should include RRIDs for each antibody [33,34] (https://scicrunch.org/resources). Unlike the company or supplier and catalog number, RRIDs do not change even if the catalog number changes or if the antibody is sold to another company or discontinued. Scientists using polyclonal antibodies should report the lot number, as there may be batch-to-batch variability due to the use of different animals for antibody production. Antibody dilution, incubation time, and temperature, as well as the total amount of protein loaded on the gel, all influence the specificity of the labeling and western blot quantification, so detailed information on each of these is essential for replication.

While shortcut citations are relatively common in western blot methods sections, they are problematic for reproducibility because it is not always possible to ascertain which methodological details in the cited paper are meant or the citation may not adequately describe the method [35]. Authors who use shortcut citations should ensure that the cited source describes methods in enough detail to replicate the experiment and that the methods used in the cited source closely correspond to their own methods. They should carefully note which elements of the cited method were used in the current study and describe any deviations from it. An alternative approach is to deposit detailed western blot protocols in an open access protocol repository (e.g., protocols.io, Protocol Exchange) and cite these protocols in the methods section. Box 2 provides more information on how to write a replicable western blot methods section.

## Implications for journal policies

Our results provide an overview of common suboptimal practices that could be targeted in submission guidelines and publication policies to improve western blot image and data presentation and methods reporting. Many journal guidelines and policies currently lack such specific recommendations. All journals should routinely request full-length western blot source data prior to publication to ensure these data exist [28]. Ideally, these data should be available to readers once the paper is published. Source data can help editors, reviewers, and authors to address questions about potential errors or misconduct (e.g., duplicated images) [37]. While there are clear advantages to requiring full blots, doing so would place labs with limited resources at a disadvantage. As a compromise, journals could ask authors to provide at least 1 full-length blot for each western blot assay. Unfortunately, previous work on policies related to RRIDs and reporting guidelines has suggested that changing journal guidelines has a limited impact on the quality of reporting [33,38–40]. An automated tool designed to screen

## Box 2. Common vs. good examples of western blot methods reporting

### Use methodological shortcut citations responsibly

**Common statement:** Western blotting was performed as described previously (citation of previous paper).

**Good example 1:** A detailed protocol of western blotting procedures used in this study is available at dx.doi.org/10.17504/protocols.io.81wgb6z2olpk/v2 [36].

**Good example 2:** Western blotting for human haptoglobin was performed using the sample preparation, blocking and antibody incubation protocols, and primary and secondary antibodies described previously (citation of open access paper that describes the method in detail). In contrast to the previous paper, the load volume was increased to 2 μl of serum.

### How much protein was loaded on the gel?

**Common statements:** 15–80 μg of protein was loaded on the gel. An equal amount of protein was loaded in each lane.

**Good example:** 15 μg of total protein was loaded to detect protein A. 80 μg of total protein was loaded to detect protein B.

### Membrane blocking protocol

**Common statement:** After blocking, membranes were incubated with primary antibodies.

**Good example:** Membranes were blocked in 5% milk in Tris-buffered saline with Tween20 (TBS-T; 20 mM Tris, 150 mM NaCl, 0.1% Tween20) for 1 h at room temperature.

### Antibody identifiers and labeling protocol

**Common statement:** The membranes were probed with anti-TurboGFP antibodies (Evrogen) followed by incubation with appropriate secondary antibodies.

**Good example:** The membranes were probed with anti-TurboGFP antibodies (1:1,000, Evrogen, cat nr #AB513, lot nr #51301010912, RRID: AB_20544089) overnight at 4 degrees. The blots were then incubated with anti-rabbit HRP-conjugated secondary antibodies (1:5,000, Innovative Research, cat nr #IGAR-HRP, lot nr #150943, RRID: AB_11041560) for 1 h at room temperature.

**Recommendation:** Report antibodies in a table to present all information in an easily accessible manner. Include columns for application (e.g., western blot, immunohistochemistry), antibody target (with conjugation if any), company or supplier, catalog number, lot number, RRID, and dilution. Note which secondary antibody was used for each primary antibody. See S3 Table for a template. For writing an informative western blot methods section, we encourage authors to use Gilda's reporting template [4].

blot images, or assess western blot methods, for factors outlined in this article and in previous guidelines could potentially help journals to screen submitted manuscripts for common problems. Furthermore, journals can ensure that all raw blots are available to readers and that all blots have molecular weight markers and labels, as well as provide a template to help authors generate informative western blot methods section [4] and a template for reporting antibodies (S3 Table).

### Additional strategies and resources for improving western blots

While this paper addressed several practices related to the display of western blot images, data presentation, and methods reporting, there are many more important practices that were beyond the scope of the current study. Proper labeling of spliced blots, for example, is critically important to show readers how the blot presented in the paper differs from the source image [9]. It's also important to ensure that the blot background is gray, rather than white. While scientists are permitted to lighten or darken blots, these adjustments should be uniform across the entire blot [41]. Once any section of the blot background is white, or maximally lightened, any further lightening would be nonuniform. Excessive lightening can be used to reduce background (Fig 6, panel B) and, in highly problematic cases, to eliminate faint bands (Fig 6, panel C). A white background also makes it easier to conceal other potentially problematic

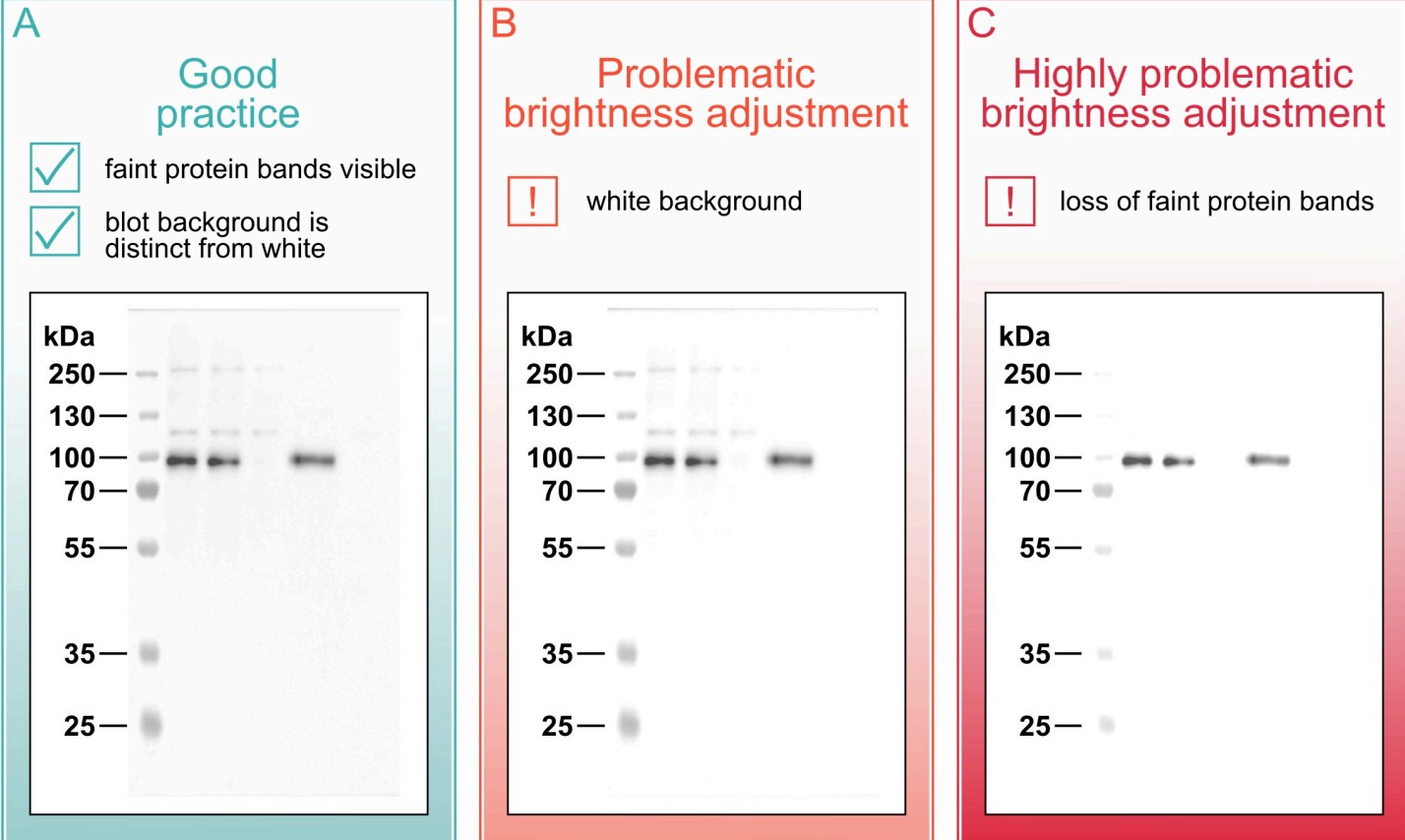

**Fig 6. Ensure that the blot background is gray.** Good practice: The blot background should be clearly distinguishable from the white page. A gray background provides assurances that the blot has not been lightened excessively. Problematic brightness adjustment: Once any section of the blot background is white, uniform lightening of the blot is no longer possible as the white region is already maximally lightened. Highly problematic brightness adjustment: In extreme cases, excessive nonuniform brightening of the blot can be used to eliminate faint bands. For the unprocessed image, see S1 Fig.

**Table 1. Resources to help scientists make western blot experiments more informative and reproducible.**

| Frequently Asked Questions | Reference | Highlights |
|---|---|---|
| **Where can I find a recent, detailed overview of western blotting?** | | |
| Overall | [42] | Image display recommendations |
| Semiquantitative blotting | [1] | Methodological reporting and data presentation recommendations |
| **Which choices are known to influence western blot outcomes?** | | |
| Selection of antibodies | [4] | Minimal methodological reporting standards; reporting template |
| Dilution of antibodies | [4] [14] | |
| Selection of buffers | [4] | |
| Total protein loaded | [4] [14] [6] | Quality control checks for western blot quantification [6] |
| Loading control | [14] | |
| Densitometry | [13] | Western blot quantification recommendations |
| Nonlinearity error | [6] | |
| Normalizing data | [6] [43] | Tool for selecting normalization strategy |
| **How do I validate antibodies for western blotting?** | [44] | Benchmarks for user validation of antibody performance, validation results reporting |
| **How should I present my western blot quantification data?** | [15,32] | List of sources of code for dot plots [15]; excel templates for dot plots [32] |
| **What is image manipulation? How can I avoid it?** | [41] | Expert opinion with specific guidelines and visual examples |
| **Reporting replicates: what constitutes N in my experiments?** | [25] | |

adjustments, like improper splicing. Presenting blots with a light gray background that is clearly distinguishable from the white page allows readers to confirm that excessive lightening was not performed. Table 1 provides an overview of additional resources to help scientists improve western blot methods and reporting, including aspects not discussed in this paper.

## Implications for future research

This study highlights several opportunities for future research. Few papers deposited source data for western blots in supplemental files; hence, our sample size was not large enough to reliably assess features of the source data. Future studies focusing on source data may provide more insight into experimental practices (e.g., How often are investigators using molecular weight markers?). Our observations suggest that antibody lists in some publications are incomplete when compared to figures in the main text. In other publications, the western blotting technique, although evidently performed, is not described in the methods section. Future studies could assess the prevalence of these shortcomings. Our evaluations required manual review; hence, the study was limited to 2 fields. An automated screening tool would allow us to examine more fields, assess changes in practices over time, and evaluate the impact of journal policies. This would help researchers to identify fields that are most likely to benefit from interventions to improve western blot reporting and image display. The creation of an automated tool would also allow us to determine whether screening preprints or submitted manuscripts, and sharing reports with authors, improves reporting [45]. Finally, future studies could investigate aspects of western blot experiments that were not evaluated in this study but are critically important for reproducibility, such as procedures for antibody validation or semiquantification of protein bands. Future studies could also assess the prevalence of questionable image preparation practices, such as splicing [9], which were not addressed in this study.

## Limitations

This study is exploratory and confirmatory studies are needed. While the protocol was established prior to data collection, it was not preregistered. Changes to the protocol have been

described in the methods section. We relied on the metadata in 1 database (Dimensions); therefore, we cannot exclude the possibility of database bias. We only examined full-length original research papers with western blot images in the main paper. Papers that reported the results of a western blot experiment without including an image were excluded. Our protocol could not differentiate whether blots were physically cut during the experiment or digitally cropped after imaging. Scientists sometimes cut blots to conserve antibodies. In many cases, both cutting and cropping may have occurred. Our results may overestimate how well anti-bodies were reported, as our protocol was designed to capture how detailed the antibody iden-tifiers were for antibodies that were reported. Reviewers did not compare figures with methods to confirm that antibodies for all blots shown in figures were also listed in the methods.

## Conclusions

Our meta-research study of common western blotting publication practices in neuroscience and cell biology highlights several shortcomings that can prevent readers from critically assess-ing or independently replicating western blot experiments. Most papers with western blot images in our sample omitted information in image display, data presentation, and methods reporting. These observations support previous calls to improve western blot reporting [4,10,11] and provide further insight into which practices urgently require attention. We uti-lized these findings to develop targeted recommendations for improving the quality of western blot reporting in scientific publications with the interrelated goals of increasing rigor and reproducibility. We have deposited a set of slides in English and Spanish that can be used to teach best practice to optimize western blots in scientific publications.

## Materials and methods

This manuscript was prepared using guidance from the Strengthening the Reporting of Obser-vational Studies in Epidemiology (STROBE; RRID:SCR_018788) reporting guidelines for observational studies and relevant items from the Preferred Reporting Items for Systematic Reviews and Meta-Analyses guidelines (PRISMA; RRID: SCR_018721) [46,47]. Ethical approval was not required. A description of western blotting materials and methods for the blot in Figs 1, 2, 5 and 6 is available on protocols.io (RRID:SCR_010490) at dx.doi.org/10.17504/protocols.io.81wgb6z2olpk/v2 [36].

### Data sources

JCR (Clarivate; RRID:SCR_017656) generates journal impact factors along with other citation data from the journals and conference proceedings in the sciences and social sciences indexed in Web of Science (Clarivate). Lists of all journals assigned to the JCR categories "neurosci-ences" ($n = 293$) and "cell biology" ($n = 201$) were downloaded on August 13, 2021. The jour-nals in each category were ranked according to their 2020 Journal Impact Factor (JIF), and the top 25% of journals from each category were selected for further analysis ($n = 68$ in neurosci-ences, $n = 49$ in cell biology). No journals were in the top 25% of both categories. Journals that did not publish original research articles were excluded via manual screening by 2 independent reviewers (FB, AK, LJ, FRUG). Discrepancies were resolved by consensus-based discussion. In total, 92 journals were included, 55 from neurosciences and 37 from cell biology. The Interna-tional Standard Serial Numbers (ISSNs) were identified for each of the included journals and used to generate a search strategy for Dimensions (NR). Lists of included journals by category are available in the Supporting information (S1 and S2 Tables).

Dimensions (Digital Science; RRID:SCR_021977) is a scholarly search database that aggregates, harmonizes, and links records of various publication types from diverse sources, including Crossref and PubMed Central, using automated routines and artificial intelligence technologies [48]. We have institutional access to the paid version and searched via API. We searched by ISSN and Dimensions date field for articles published in each of the included journals with an earliest publication date (either print or electronic) between July 1 to 31, 2021. The search was run on October 12, 2021. Search results were not compared against another source. The search date was selected to account for possible lags in indexing, publishing, and aggregating records. The search returned a total of 2,426 articles, including 1,386 in the neurosciences journals and 1,040 in the cell biology journals.

## Article selection

**Automated screening to identify papers with blots or gels.** PDFs of all full-text articles identified by the search were screened by an open-source automated tool designed to identify selected image types, including blots and gels, in biomedical publications. The tool has high sensitivity (94%), precision (98%), and F1-scores (0.96) for identifying pages that include a blot or gel. The tool runs on fastai (https://docs.fast.ai/), a library built on PyTorch, and uses resnet101, a well-established pretrained deep convolutional neural network. The image resolution was set to $560 \times 560$ pixels to detect small blots. Code and further information, including additional performance metrics, can be accessed on GitHub (https://github.com/quest-bih/image_screening_tool)) [49].

The automated screening tool excluded 1,800 papers, of which 586 papers were from cell biology journals and 1,215 papers from neuroscience journals (S2 Fig). To confirm that the screening tool was effective, a random sample of 5% of the automatically excluded documents per JCR category ($n = 30$ neurosciences; $n = 61$ cell biology) was manually validated by 2 independent reviewers (FU and CM). Discrepancies were resolved by consensus-based discussion. None of the papers in the validation set contained western blot images. The automated screening tool identified 625 papers that contained a blot or gel. These were manually screened for eligibility as described below.

**Procurement of full-texts and supplementary materials.** A total of 625 PDFs of full-text articles were procured, including 171 from the neurosciences journals and 454 from the cell biology journals. A total of 554 papers (neuroscience: n = 147, cell biology: n = 417) had supplements, all of which were procured. Additional full-text articles of 91 papers that the screening tool identified as having no gels or blots were obtained to validate the automated screening tool results.

**Manual screening to identify papers with western blots.** Papers containing a blot or gel, as determined by the automated screening tool, were reviewed by two independent reviewers in rotating pairs (AA, EA, LB, FB, MF, BG, MH, LJ, VK, AK, CK, CM, LS, FRUG). The screening protocol consisted of 2 items that determined the inclusion or exclusion of the paper and a reason for exclusion (see abstraction protocols). Papers were excluded if they did not contain western blots or were not full-length original research. In total, 551 papers included western blots and were eligible for abstraction (neuroscience: n = 151, cell biology: n = 400; S2 Fig).

## Data abstraction

Each reviewer completed a training set of 30 papers and a predefined consensus rate of $\geq 80\%$ was met by each team before starting data abstraction. Data abstraction was performed according to 2 protocols (figures protocol and methods protocol) by 2 specialized teams of independent reviewers working in rotating pairs. One team abstracted 13 items relating to the display

of western blot images in figures and the presentation of western blot quantification data from figures, figure legends, figure descriptions, and supplements (figures team: AA, EA, FB, MF, MH, LJ, AK, CM, FRUG). A second team abstracted 19 items relating to the reporting of western blot methods and materials from the Materials and methods section and Supporting information (methods team: LB, BG, CK, LS, VK). Discrepancies were resolved by consensus-based discussion.

The protocols were established before data abstraction began but were not preregistered. One question related to reporting of replicates was eliminated from both protocols during data abstraction (Box 1). The full protocols are available in the OSF repository at https://osf.io/2c4wq [24].

## Domains of figures protocol

1. **Cropping:** Are the blots in the paper cropped?

2. **Molecular weight markers:** Do the blots have a molecular weight marker?

3. **Molecular weight labels:** Do the blots have molecular weight label(s) and what is the position of the(se) label(s)?

4. **Blot source data:** Is western blot source data provided in the supplement? If yes, is it a full-length blot and does it have a molecular weight marker?

5. **Quantification graphs**: Is there a western blot–associated graph? If yes, what type of graph is it?

The edges of the gel and loading lanes were rarely shown; therefore, western blots were defined as full length based on the range of molecular weight labels or number of visible marker bands (≥approximately 8 maker bands were considered sufficiently full length). The range of molecular weights shown depends on the gel percentage; therefore, it was not possible to set strict criteria for molecular weight labels. If the highest label was closer to 100, we would expect the lowest label on a full gel to be closer to 10. If the highest label was several hundred kDa, the lowest label might be closer to 100. We are confident that all blots classified as full blots included most of the blot but cannot exclude the possibility that some very high or very low bands from the molecular weight marker were excluded.

## Domains of methods reporting protocol

1. **Methodological shortcut citations:** Is there a shortcut citation in the reporting of the western blot methods?

2. **Protein loading:** Do the authors report information on how much protein was loaded on the gel? Is the amount reported as an exact number, a range of values, or a combination of both?

3. **Membrane blocking protocol:** Do the authors report details concerning the blocking reagent, concentration, and duration?

4. **Primary and secondary antibodies:** Do the authors report the antibody company or supplier, catalog number, lot number, RRID, dilution, duration, and temperature of incubation for primary and secondary antibodies?

## Data verification

We checked for consistency across abstraction pairs in a random selection of 10% ($n = 62$) of papers from the sample. Verification abstraction for figures was performed by a single abstractor from the methods team (CK). Verification abstraction for the methods team was performed by all members of this team (LB, BG, CK, LS, VK). Papers were assigned such that each member verified papers that they had not originally abstracted. Results of both verification abstractions were compared to the consensus from the original abstraction. Discrepancies were resolved by group discussion. Error rates were calculated as percentage of ratings for which the original abstractors' response was incorrect. Error rates were 5.5% for the figures abstraction and 1.4% for the methods abstraction protocols.

## Data processing and creation of figures

Data are presented as n (%) and were processed and analyzed using Python Programming Language (RRID:SCR_008394, version 3.8.8, libraries NumPy 1.19.5 and Matplotlib 3.5.1 and Pandas 1.2.4). Data visualization for Figs 3 and 4 were created using Python-based Jupyter Notebooks (RRID:SCR_018315, version 6.3.0). Code is available in our repository [24]. Each panel of Fig 3 was compiled separately and combined with the vector graphic software Adobe Illustrator (RRID:SCR_010279, version 26.0.3). An example western blot image was generously provided by one of the authors and processed, cropped, and annotated to prepare "mock examples" for Fig 1 (created with BioRender.com, RRID:SCR_018361), Figs 2, 5 and 6 (created with the vector graphic software Inkscape, RRID:SCR_014479, version 1.0.2–2). All suboptimal western blot images presented were created to highlight practices observed during data abstraction. Figures were tested for clarity with the colorblindness simulation tool Color Oracle (RRID:SCR_018400).

## Supporting information

**S1 Fig. Raw image.** Original, uncropped, and unprocessed image supporting Figs 1, 2, 5 and 6. (DOCX)

**S2 Fig. Modified PRISMA 2020 flowchart of screening and selection process.** This flowchart depicts the screening and selection process, including the number of journals and articles excluded and the reasons for exclusion at each stage of the study. (DOCX)

**S1 Table. Number of articles examined by journal (neurosciences).** Values are n, or n (% of all articles). Articles that were not full-length original research articles (reviews, editorials, perspectives, commentaries, letters to the editor, short communications, etc.) or did not include eligible images were excluded. (DOCX)

**S2 Table. Number of articles examined by journal (cell biology).** Values are n, or n (% of all articles). Articles that were not full-length original research articles (reviews, editorials, perspectives, commentaries, letters to the editor, short communications, etc.) or did not include eligible images were excluded. (DOCX)

**S3 Table. Antibody reporting template.** A tabular antibody reporting template. (DOCX)

## Acknowledgments

We thank Camila Victoria-Quilla Baselly Heinrich for her help with data curation and course administration and Joachim Fuchs for helpful comments and feedback on the manuscript.

## Author Contributions

**Conceptualization:** Cristina Kroon, Larissa Breuer, Lydia Jones, Felix Busch, Biswajit Ghosh, Vartan Kazezian, Nico Riedel, Lea Scherschinski, Tracey L. Weissgerber.

**Data curation:** Jeehye An, Ayça Akan, Cesar Alberto Moreira Restrepo.

**Formal analysis:** Jeehye An.

**Funding acquisition:** Tracey L. Weissgerber.

**Investigation:** Cristina Kroon, Larissa Breuer, Lydia Jones, Ayça Akan, Elkhansa Ahmed Mohamed Ali, Felix Busch, Marinus Fislage, Biswajit Ghosh, Max Hellrigel-Holderbaum, Vartan Kazezian, Alina Koppold, Cesar Alberto Moreira Restrepo, Nico Riedel, Lea Scherschinski, Fernando Raúl Urrutia Gonzalez.

**Methodology:** Cristina Kroon, Larissa Breuer, Lydia Jones, Ayça Akan, Felix Busch, Biswajit Ghosh, Max Hellrigel-Holderbaum, Vartan Kazezian, Lea Scherschinski, Tracey L. Weissgerber.

**Project administration:** Cristina Kroon, Larissa Breuer, Jeehye An, Ayça Akan, Tracey L. Weissgerber.

**Resources:** Cristina Kroon, Nico Riedel, Tracey L. Weissgerber.

**Software:** Nico Riedel.

**Supervision:** Cristina Kroon, Larissa Breuer, Tracey L. Weissgerber.

**Validation:** Cristina Kroon, Larissa Breuer, Ayça Akan, Biswajit Ghosh, Vartan Kazezian, Alina Koppold, Cesar Alberto Moreira Restrepo, Lea Scherschinski, Fernando Raúl Urrutia Gonzalez.

**Visualization:** Larissa Breuer, Jeehye An, Alina Koppold, Tracey L. Weissgerber.

**Writing – original draft:** Cristina Kroon, Larissa Breuer, Lydia Jones, Marinus Fislage, Lea Scherschinski.

**Writing – review & editing:** Cristina Kroon, Larissa Breuer, Lydia Jones, Jeehye An, Ayça Akan, Elkhansa Ahmed Mohamed Ali, Felix Busch, Marinus Fislage, Biswajit Ghosh, Max Hellrigel-Holderbaum, Vartan Kazezian, Alina Koppold, Cesar Alberto Moreira Restrepo, Nico Riedel, Lea Scherschinski, Fernando Raúl Urrutia Gonzalez, Tracey L. Weissgerber.

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
