## [Editor Report · Decision Letter 0]

15 Jun 2022

Dear Tracey, 

Thank you for submitting your manuscript entitled "Blind spots on western blots: a meta-research study highlighting opportunities to improve figures and methods reporting" for consideration as a Meta-Research Article by PLOS Biology.

Your manuscript has now been evaluated by the PLOS Biology editorial staff, and I'm writing to let you know that we would like to send your submission out for external peer review.

Once your full submission is complete, your paper will undergo a series of checks in preparation for peer review. After your manuscript has passed the checks it will be sent out for review. To provide the metadata for your submission, please Login to Editorial Manager (https://www.editorialmanager.com/pbiology) within two working days, i.e. by Jun 17 2022 11:59PM.

Kind regards,

Roli

Roland Roberts, PhD

Senior Editor

PLOS Biology

rroberts@plos.org

---

## [Decision Letter · Decision Letter 1]

15 Jul 2022

Dear Tracey,

Thank you for your patience while your manuscript "Blind spots on western blots: a meta-research study highlighting opportunities to improve figures and methods reporting" was peer-reviewed at PLOS Biology. It has now been evaluated by the PLOS Biology editors, an Academic Editor with relevant expertise, and by two independent reviewers. 

Based on the reviews, we are likely to accept this manuscript for publication, provided you satisfactorily address the remaining points raised by the reviewers. Please also make sure to address the following data and other policy-related requests.

IMPORTANT:

a) Please attend to the requests from the reviewers.

b) Please change the title to something more informative and explicit; we suggest "Assessment of common problems in western blot figures and guidelines to make them informative, reproducible and transparent"

c) Please supply a blurb according to the instructions in the submission form.

d) Please cite the location of the data clearly in the legends to Figs 3 and 4, e.g. “The data underlying this Figure can be found at https://osf.io/2c4wq/”

We expect to receive your revised manuscript within two weeks. 

*Published Peer Review History*

*Press*

Sincerely,

Roli

Roland Roberts, PhD

Senior Editor,

rroberts@plos.org,

PLOS Biology

DATA NOT SHOWN?

REVIEWERS' COMMENTS:

Reviewer #1:

[identifies herself as Elisabeth M Bik]

In this manuscript, the authors carefully examined Western blots in a set of 551 articles published in top journals in the fields of neurosciences and cell biology, and scored practices such as the presence of marker lanes or marker labels, original, uncropped blots, and detailed description about blocking steps and antibodies used. Cropped blots without originals were very common, and details or labels were partially or completely missing in most of the papers. The authors provide detailed numbers on such practices, as well as guidelines to help researchers prepare figures and methods sections involving Western blots.

The manuscript is very clearly written, and the figures and textboxes are well designed and easy to understand. The article is also timely and much needed. As research papers get more complicated, details needed to replicate an experiment are often not included. Scientific journals, publishers, and peer reviewers seem to often not check if such details are included. It would be recommended if the guidelines provided in this manuscript could e.g. be included in submission guidelines or 'instructions for authors' found at scientific journals.

I enjoyed reading this manuscript and have only minor comments. 

 1 General question: How did the authors define the difference between a full length blot and a cropped one? E.g. did the blot have to show edges of the membrane / film in order to be called full length, or was showing a range of e.g. 10 to 100 kDa considered full length? Some more details on the authors' definition for this study might be helpful.

 2 General question: Did the authors look for or observe any splicing in Western blots? Splicing is another questionable research practice that might deserve to be briefly mentioned. There is a very quick mention on Page 11 ('Annotations should also note lanes that appear on the original blot but were cropped out of the region shown in the figure'). Perhaps one or two lines about splicing could be added to e.g. the discussion section. Fosang et al. can be used as a reference, since it is already used in this manuscript (reference 10).

 3 Page 5: Introduction: textbox 3 - no textbox 1 and 2 were mentioned before this one. Is the numbering correct? Or should this be 'textbook 1'?

 4 Figure 3: This is minor, but perhaps relevant in a paper that discusses good practices on figure information: The figure has some inconsistencies in the labeling. In Section A, the number of papers is given in the text under the bars "% of papers (n=151)", while in Section B and C, it is given in the text above the bars "Neuroscience (n=151)". 

 5 Page 8: 'Membrane blocking protocol: The membrane blocking protocol was rarely reported in detail' - I am not sure if 'rarely' is a good word here. Looking at Figure 4C, it seems that one third to half of the papers do this, so in comparison to other findings in this study, that is actually pretty good. 

 6 Page 9: What exactly did the authors mean by papers reporting on the source of primary antibodies? Does it refer to the name of the manufacturer, or the animal species in which it was generated?  The word 'source' is also used in other parts of this manuscript to describe original, uncropped blots, so it would be good to give a precise definition of the word 'source' in this particular context here. 

 7 Page 9, box "Reporting Replicates". For biological replicates, the authors write "e.g. 5 patients or mice" but I am not sure if samples from 5 patients could be considered biological replicates, since they are from different individuals. Rather, one might think of e.g. 5 blood or stool samples from the same patient, or even different aliquots from the same stool sample. For inbred mice kept at the same conditions, one might argue that these are biological replicates, but humans have much more variation. See e.g. https://www.nature.com/articles/nmeth.3091 But I will admit that I am not sure about the definition, and I will leave it up to the authors to decide if this wording can stay.

 8 Page 9, box "Reporting Replicates": it might be better to change the word "source" in "Technical replicates are not independent as these samples originate from the same source" to "specimen", since the word 'source' is also used to describe other things in this study.

 9 Page 10. "Some journals and guidelines [11, 12, 23] require authors to submit original uncropped images of blots".  Were any of the journals included in this study journals that required authors to submit uncropped blots (e.g. in the supplemental materials)? If so, did this study still find that such originals were missing? As a suggestion for an additional analysis on this dataset, the authors could add such findings to e.g. Figure 3A, or they could make a brief statement about their findings. Did those journals requiring original blots do better in the analysis here? This additional analysis is just a suggestion, not a requirement on my behalf for acceptance of this manuscript. 

 10 Page 12, Figure 5. "Published western blots should show a minimum of two molecular weight markers" - maybe word this a bit more precisely. The authors probably do not mean a minimum of two lanes with markers, but a minimum of two bands of different weight. 

Signed, Elisabeth Bik

Reviewer #2:

The authors inspected Western blots in 551 original research articles across 92 journals, 55 from neurosciences and 37 from cell biology. The top 25% of journals from each category were selected. 

* most published Western blots are cropped and blot source data are not made available to readers in the supplement. 

* publishing blots with visible molecular weight markers is rare, and many blots additionally lack molecular weight labels. 

* Western blot methods sections often lack information on the amount of protein loaded on the gel, blocking steps and antibody labeling protocol. 

* Important antibody identifiers like source, catalog number or RRID were omitted frequently for primary antibodies, and regularly for secondary antibodies. 

Their data reveal a significant lack of attention by journals in this regard, and points to a need for improved reporting standards, and more specific recommendations in journal guidelines.

The article is well written and the figures are clear and informative. The authors also include a toolbox to help scientists produce more reproducible western blot data, teaching slides in English and Spanish, and an antibody reporting template. 

The authors describe common shortcomings in methods reporting, data visualization, and image display. Addressing these would significantly improve assessability, reproducibility and trustworthiness of the research reported. 

Suggestions:

* The authors mention that proper reporting 'may come at a cost ' in the Introduction, but do not discuss this much further. The cost is presumably mainly monetary (e.g. cutting blots to save antibodies), although there may also be a time factor. It appears that blotting techniques differ around the world, as do standards of imaging and how the raw images are prepared and stored. Geographic differences are not mentioned or discussed.

* The authors briefly mention image integrity. They could point out that full information and complete raw data are crucial during investigations addressing questions regarding potential misconduct issues or errors (experimental or during figure preparation). Journals should routinely request to see the original and full raw Western blot images prior to publication, even if they aren't published alongside the research articles in the end. 

* Over-exposure or over-contrasting of the final images, resulting in a lack of background noise is not mentioned although this is a common problem. Over-contrasting can be used to hide weaker signals/bands, to hide splice marks and cuts as well as a variety of other digital image manipulation issues.

* Regarding cropping: It might be worth explaining that in some countries it appears to be commonplace to cut membranes into slim strips before they are placed into an imaging device. Instead of blots displaying the entire vertical length of the gel, these strips are often the only raw data available and typically just show the band of interest, frequently also without showing molecular weight markers and -labels. This also makes it difficult to verify whether controls were run on the same blot as the proteins of interest. If confronted, authors often explain that this is common practice to cut costs. Some journals appear to still accept this type of raw data despite the clear drawbacks.

The results of the present study reveal a lack of fastidiousness by some journals and could be an important contribution enforcing improved reporting standards.

---

## [Editor Report · Decision Letter 2]

4 Aug 2022

Dear Tracey,

Thank you for the submission of your revised Meta-Research Article "Blind spots on western blots: assessment of common problems in western blot figures and methods reporting with recommendations to improve them" for publication in PLOS Biology. On behalf of my colleagues and the Academic Editor, Mathieu Bertrand, I'm pleased to say that we can in principle accept your manuscript for publication, provided you address any remaining formatting and reporting issues. These will be detailed in an email you should receive within 2-3 business days from our colleagues in the journal operations team; no action is required from you until then. Please note that we will not be able to formally accept your manuscript and schedule it for publication until you have completed any requested changes.

Best wishes, 

Roli

Senior Editor

PLOS Biology

rroberts@plos.org